# The Functional States of the Participants of a Marine Arctic Expedition with Different Levels of Vitamin D in Blood

**DOI:** 10.3390/ijerph20126092

**Published:** 2023-06-09

**Authors:** Natalia Simonova, Maria Kirichek, Anna A. Trofimova, Yana Korneeva, Anna N. Trofimova, Rimma Korobitsyna, Tatiana Sorokina

**Affiliations:** 1Laboratory of Labor Psychology, Faculty of Psychology, Moscow State University Named after M.V. Lomonosov, Moscow 125009, Russia; n.simonova@narfu.ru; 2Department of Psychology, Northern (Arctic) Federal University Named after M.V. Lomonosov, Arkhangelsk 163002, Russia; m.tunkina@narfu.ru (M.K.); trofimova.a.a@edu.narfu.ru (A.A.T.); a.trofimova@narfu.ru (A.N.T.); r.korobicina@narfu.ru (R.K.); t.sorokina@narfu.ru (T.S.)

**Keywords:** functional state of a person, stress, working capacity, vitamin D, dynamic monitoring, adaptation, the Arctic

## Abstract

(1) Background: The vitamin D level in blood is one of the markers of the functional reserves of the human body and can contribute to more successful adaptation in the Arctic. (2) Methods: The study involved 38 participants in the project “Arctic Floating University—2021”. The determination of vitamin D content was carried out at the beginning of the expedition. A dynamic study was carried out for 20 days in the morning and in the evening. The functional state parameters of the participants were assessed using psychophysiological and questionnaire methods. Statistical methods: Mann–Whitney U-test and correlation analysis. (3) Results: It was found that at the beginning of the expedition, the functional state of participants with more severe vitamin D deficiency is characterized by a shorter average duration of RR intervals (*p* = 0.050) and reduced SDNN values (*p* = 0.015). The higher the content of vitamin D, the greater increase in speed (*r* = 0.510), the higher the increase in projective performance (*r* = 0.485), and the smaller the increase in projective stress (*r* = −0.334). Significant relationships between the subjective characteristics of functional states and the vitamin D of participants have not been established. (4) Conclusion: With an increase in the severity of vitamin D deficiency in the blood, the adaptive capabilities of participants decrease during an expedition to the Arctic.

## 1. Introduction

Adaptation and human activity in the Arctic have been the subject of many studies. This fact can be explained by the extreme nature of Arctic climatic and geographical conditions, including low temperatures combined with intense winds and high humidity, sudden changes in barometric pressure, polar day and polar night, etc. [1,2,3]. These studies have gained particular relevance in the context of climate change in the Arctic and its consequences for humans and their health [4,5,6,7,8,9,10]. In this context, research into the impact of climate change in the Arctic and adaptation to it is of great importance for practitioners in the field of labor protection and management of enterprises operating in the Arctic.

The results of polar medicine research [11,12] reflect the importance of monitoring the functional (psycho-physiological) states of people living and carrying out professional activities in the extreme conditions of the Arctic region and subarctic territories. Adaptation to adverse climatic and geographical conditions and the maintenance of high working capacity are associated with an increase in the intensity of energy costs and metabolic and other costs to the body, and hence an increase in the need for a sufficient amount of functional reserves and their rapid mobilization.

The polysyndrome of depletion of the body’s adaptive reserves under the influence of factors at high latitudes is called polar stress syndrome, the main component of which is oxidative stress [11]. According to V.I. Khasnulin [11] and his colleagues [12], who studied these problems for many years, the specificity of this syndrome is due to the fact that in the North the effect of negative environmental factors on the human body can be traced in a sequence different from the sequence of the adaptation syndrome development formulated by G. Selye [13,14]. Thus, adaptive and disadaptive processes in the North begin not in the central nervous system (as is commonly believed), but at the molecular–cellular level of the organism. Gradually, malfunctioning of internal organs develops, which then leads to an imbalance in the work of the endocrine glands, blood vessels, heart, and other life-supporting systems. Psycho-emotional tension completes the picture of changes in this case [11].

The features of adaptation to the extreme natural and climatic conditions of the Arctic can be observed when studying the dynamics of the state of participants in scientific expeditions. A study by Bulgarian scientists found that participants’ perceived stress levels at the beginning of an Antarctic expedition were significantly higher compared to the results at the end of the expedition [15]. A high level of perceived stress has the greatest impact on anxiety levels in study participants [15]. The dynamics of stress and recovery reactions and their relationship with the perception of environmental development during one year of polar wintering of expedition members in various environmental conditions of the subantarctic and Antarctic polar stations was studied by the international scientific team [16]. The studies showed that stress and recovery reactions of winterers were characterized by different dynamics depending on the degrees of environmental extremeness (the authors found that the stress response duration is proportional to the severity of environmental conditions: it is longer in polar conditions than in subantarctic ones). [16]. The features of the cardiovascular system in participants of high-latitude expeditions have been established by Russian scientists as markers of positive and physiologically adequate adaptation shifts in the autonomic regulation of the cardiovascular system [17]. Multiple stressors play quite a big role in adaptation to climate change in the Canadian Arctic [18]. Effective coping strategies used during the extreme Antarctic expedition were described in Reference [19].

According to ongoing studies [20], at least half of the planet’s population has some degree of vitamin D deficiency; the average level in this country is only 22.4 ng/mL [21]. Holick et al. note that with increasing age, the percentage of the population with severe vitamin D deficiency also increases, reaching 80–90% [22]. The content of vitamin D in blood can also act as a marker of the functional reserves of the human body and is an important element in maintaining human health in the Arctic. Vitamin D deficiency is associated with various diseases, such as cardiovascular disease [23], metabolic syndrome, type 2 diabetes mellitus, infectious/inflammatory diseases, autoimmune diseases, and cancer [22,24]. It should be noted that vitamin D sufficiency affects the functioning and regulation of reproductive functions in both women and men [25]. The main factors affecting the availability of vitamin D include place of residence (namely geographical latitude), season of the year, insolation level, nutritional habits of the population, age, and concomitant diseases, for example, gastrointestinal ones [26].

According to R.M. Baevsky, functional reserves are understood as “... informational, energy, metabolic resources of the body, providing its specific adaptive capabilities. In order to mobilize these resources under changing environmental conditions, a certain tension of regulatory systems is necessary. It is the degree of tension of regulatory systems necessary to maintain homeostasis that determines the current functional state of a person,” [27]. Evaluation and prediction of the functional state of the whole organism according to data from study of the cardiovascular system is based on the fact that hemodynamic changes in various organs and systems occur earlier than their corresponding functional disorders. The study of the processes of temporal organization, of coordination and synchronization of information, and of energy and hemodynamic processes in the cardiovascular system makes it possible to identify the very initial changes in the control link of the whole organism. The cardiovascular system, with its regulatory apparatus, is considered to be an indicator of adaptive reactions of the whole organism; its regulation reflects all levels of control of physiological functions [27]. The cardiovascular system is the coordinating bond between the controlling and controlled links.

However, the relationship between vitamin D deficiency and successful adaptation to activities in extreme conditions has been discussed much less, which is an omission [28,29]. Unfavorable working conditions, such as shift work, irregular working hours, and a rotation system are important social issues throughout the world [30]. A prospective study conducted in the UK showed that long working hours, change of location, isolation, and lack of free time led to the development of depressive and anxiety symptoms in workers [31]. At the same time, studies in the regions of the Far North and in the North-West of Russia revealed a lack of vitamin D in military personnel [32]. Several studies have established an epidemiological relationship between blood vitamin D levels, shift work, and work patterns [33,34].

The goal of our study was to identify and describe the relationship between changes in objective psycho-physiological and subjective psychological parameters of functional states and the level of vitamin D deficiency in the blood of participants in a sea expedition to the Arctic region over the course of a 20-day trip.

In order to achieve this goal, the features of the averaged (over the entire period) indicators of functional states in the group of participants with vitamin D deficiency were determined, and the relationship between individual indicators of the dynamics of functional states and vitamin D level in the blood of the expedition participants was revealed.

Research hypotheses:We assumed that the indicators of positive characteristics of functional states in expedition participants with severe vitamin D deficiency in the blood would be significantly lower, and those of negative characteristics—higher.We expected that the lower the level of vitamin D, the worse the participants of the expedition would retain positive characteristics of functional states. At the same time, the dynamics of functional states would be most pronounced when measured by objective and projective methods.

The positive characteristics of the functional states of the expedition members were the optimal level of the general functional state of the body, operator working capacity, working capacity, favorable levels of well-being, activity, and mood. The negative characteristics of the participants’ functional states included negative levels of the general functional state of the body, the presence of a stressful state, the presence of unproductive neuromuscular activity (SO), reduced levels of well-being and activity, as well as depressed mood.

## 2. Materials and Methods

### 2.1. Sample

The dynamic study was carried out on the research expedition vessel “Mikhail Somov” for 20 days in the morning and evening from 11 to 30 June 2021. Expedition route: Arkhangelsk—Cape Zhelaniya (Novaya Zemlya)—Graham Bell Island (Franz Josef Land [FJL])—Hayes island (FJL)—Hooker island (FJL)—Arkhangelsk. The route in the Barents Sea included helicopter landings on the islands for scientific research.

The study involved 38 people (18 men and 20 women aged 20 to 72 years, average age 33.4 ± 2.1 years), including participants in the scientific and educational expedition project “Arctic Floating University-2021”, National Park “Russian Arctic”, participants in the project “Master of the Arctic” and crew members. An examination by a general practitioner was a prerequisite for participation in sea expeditions, which was meant to identify contraindications to the relevant working conditions. After the check-up, the study participants were allowed to join the expedition; therefore, all of them could be considered conditionally healthy people.

All the study participants completed questionnaires consisting of the following sections: date of birth, gender, place of residence, as well as questions about the intake and frequency of intake of bioactive food supplements containing vitamin D.

Participation in the study was voluntary. All of the participants signed an information consent form and consent for the processing of personal data. The research program and methods were reviewed by the ethics committee of the Northern State Medical University of the Ministry of Health of Russia (Arkhangelsk, Russia) and were recommended for use (protocol No. 04-06-21 dated 9 June 2021).

### 2.2. Methods

#### 2.2.1. The Biochemical Method for Assessing the Content of Vitamin D in the Blood of Expedition Participants

In the present study, vitamin D in the blood of the expedition members was measured once as an indicator of their working capacity. The collection of blood samples in order to determine the content of vitamin D was carried out at the beginning of the expedition period (the first three days of the study) by specialists with medical degrees.

The biochemical method involves testing the concentration of vitamin D in blood. Venous blood samples were taken by medical personnel in 9 mL Improvacuter vacutainers (Guangzhou, China). A whole blood sample was taken from each subject, from which, after centrifugation (3000 rpm), a blood serum sample was obtained. The serum was then transferred to 1.5 mL Ssibio cryotubes (SSIbio, Lodi, CA, USA) and frozen to −25 °C before transport. Transportation to the place of analysis was carried out in medical cooler bags at −25 °C without defrosting.

The study was performed by high working-capacity liquid chromatography with tandem mass spectrometric detection (HPLC-MS/MS). An Agilent 1200 (USA) liquid chromatograph and an AB Sciex 3200 MD (Singapore) mass spectrometer were used for the study. During the course of sample preparation, the precipitation of blood proteins and subsequent solid-phase extraction took place, which made it possible to remove all interfering components [35]. The used control material was manufactured by Recipe (Munich, Germany) for the determination of 25-OHD3 and 25-OHD2 vitamins for mass spectrometric studies LOT 1207 REF MS7080.

Vitamin D sufficiency was assessed on the basis of the following criteria: a vitamin D content less than 12 ng/mL was considered deficiency, the range of 12 to 20 ng/mL corresponded to deficient, more than 20 ng/mL corresponded to normal, and a concentration of more than 50 ng/mL was linked to potential adverse effects [36].

#### 2.2.2. Substantiation of an Integrated Approach to Assessing the Functional States of Expedition Members

Within the framework of the structural–integrative approach, the FS was understood as a relatively stable structure of updated internal funds for a certain period of time, characterizing the mechanisms of activity regulation that have developed in a particular situation and determining the effectiveness of solving problems [37].

The dynamics of functional states are the replacement of one set of reactions by another set of reactions. At the same time, we are talking not just about a set of parameters that describe the dynamics, but about trends in the nature of the relationship between them as elements of an integral structure [38]. The parameters of the cardiovascular, vegetative, and endocrine systems, which were measured in separate periods of time, stress, performance, etc., are related to the characteristics of a person’s functional states. They characterize the level of physiological reserves and the potential for the implementation of human activities. A decrease in working capacity can be considered a sign of deterioration in the functional state [38]. The working capacity dynamics allow one to see at what psycho-physiological “cost” a particular result of an activity is achieved and, in general, to assess whether an activity corresponds to a specialist’s capabilities [39].

To date, the question of the scope and choice of methods for studying functional states is debatable. As a rule, monitoring of a person’s functional state is carried out in a complex manner using two groups of methods [38,40,41,42,43]:(1)instrumental psychophysiological methods, which are objective because they evaluate changes at the level of physiological systems (for example, heart rate variability and sensorimotor reactions);(2)subjective psychological methods, which involve an assessment of the state at a subjective sensations level as well as a projective little-conscious assessment of the functional state.

Thus, an integrated approach to the assessment of functional states involves dynamic study of the characteristics of states using different methods. The use of a comprehensive assessment is especially in demand when studying the dynamics of a person’s functional state when working in the Arctic due to the specifics of northern stress (polar stress syndrome), when negative changes that occur can be recorded, but not realized at the subjective level [43].

An integrated approach to assessing the parameters of the functional state of the expedition participants involved the use of objective psychophysiological and psychological (projective and subjective–evaluative) research methods:

1. Psychophysiological methods using the UPFT-1/30 Psychophysiologist (MTD Medicom, Russia, Taganrog) apparatus for psychophysiological testing, which is a certified device (registration certificate FSR 2007/00125):

1.1. The method of variational cardiointervalometry (VCM), which helps to assess the general functional state, specifically the state of the autonomic nervous system of the expedition participants, based on the analysis of ECG parameters of the heart rhythm of the subjects [44]. During the process of recording the ECG signal, the time between adjacent RR intervals was measured (duration of 128 cardio intervals). The examination time was 5 min. Interpretation of the results was carried out in the following areas:(1)Using the characteristics of the distribution of cardiointervals, the level of tension of the regulatory mechanisms was estimated.(2)Based on the analysis of histograms and spectral analysis of the heart rhythm, the predominance of the sympathetic or parasympathetic systems in the regulation of the heart rhythm was studied [41].

Interpretation of the results was carried out on the basis of two statistical indicators of temporal analysis—the average duration of RR intervals between sinus contractions (RRNN) and the standard deviation of the duration of RR intervals between sinus contractions (SDNN)—as well as on the basis of an integral indicator reflecting the level of the functional state (UFS). The “NN” in the name of the indicators means a number of normal intervals, “normal to normal”, with the exception of extrasystoles [27]. By means of multiplicative convolution, according to the algorithm, the integral indicator functional state level (FSL) was calculated according to the parameters of the activity of the cardiovascular system (in the range of values from 0 to 5) [44].

1.2. The complex visual-motor reaction (CVMR-35) technique, which allows for assessment of a person’s level of operator working capacity according to two alternative parameters of a complex visual–motor reaction [44]. A two-color indicator (red or green) was used as a stimulator (light stimuli); the stimuli were presented sequentially. The color for presentation was selected automatically, in random order. The number of stimuli was 35. The first five stimuli were used for training and were not included in the calculation [44]. In our study, operator working capacity was understood as the working capacity under conditions of increased concentration of attention and high speed of decision-making as part of assessing the level of sensorimotor qualities. High working capacity in this case implies high-quality (error-free) and fast execution of the test task. To analyze the data of the CVMR-35 method, we used the following main integral criteria for assessing sensorimotor reactions: the level of activity quality (error-free) in the range of values from 1 to 5 points; the working capacity level in the range of values from 1 to 5 points. We also used quantitative data, such as the total number of errors and the average reaction time (ms).

2. The psychological method was used to assess the characteristics of the functional state based on the self-reporting of participants as well as to assess the state of the autonomic nervous system, stress, and working capacity of the participants based on the questionnaire. The projective method was used as an alternative to hardware research methods.

2.1. M. Luscher’s color test [45,46] was utilized using interpretation coefficients developed by G.A. Aminev [47] for this technique. This test belongs to the group of projective methods. In order to analyze the current state of the participants based on the color choices of M. Luscher’s test, interpretative coefficients developed by G.A. Aminev were used [47]. The author–developer calculated six key coefficients reflecting various types and manifestations of functional states by using formulas that take into account a certain combination of color choices and using factor analysis. These include performance, stress, heteronomy, balance of personality traits, concentricity, and balance of the autonomic nervous system.

The methodology for working coefficients was based on Walneffer‘s research [48]. In our previous study, their calculations, rationale, and practical significance were presented [43,49,50].

2.2. The questionnaire used for the self-assessment of states—“Well-being. Activity. Mood” (WAM) [51]—was developed by V.A. Doskin, N.A. Lavrentieva, V.B. Sharay, and M.P. Miroshnikov. The methodology was based on self-reporting by the studied participants according to three parameters (well-being, activity, and mood) which characterize the state in a specific period of time. This methodology considered 30 pairs of characteristics that were opposite in meaning (for example, “Feeling good–Feeling bad”, “Passive–Active”, and “Good mood–Bad mood”). The subjects were asked to correlate their state during a specific period of time according to a number of characteristics on a multi-stage scale (3 2 1 0 1 2 3) which was located between 30 pairs of these characteristics. Well-being was understood as a set of subjective characteristics regarding health, strength, endurance, or fatigue and reflecting the degree of physiological and psychological comfort of a person’s condition. Activity was one of the areas of manifestation of a temperament, and it was characterized by mobility, speed, and pace of functions as well as the intensity and volume of human interaction with the physical and social environment. Mood referred to a set of characteristics of the emotional state. The questionnaire’s construct validity was established on the basis of comparison with the results of psychophysiological methods, taking into account the indicators of the flashing critical frequency, body temperature, and chronoreflexometry (speed of motor reactions) [51]. The questionnaire’s current validity was established by comparing the data of contrast groups as well as by comparing the results of the subjects at different working hours [51].

Table 1 presents the positive and negative characteristics of the functional states of the expedition members, which were evaluated and analyzed in the study.

### 2.3. Procedure

The collection of blood samples in order to determine the content of vitamin D was carried out at the beginning of the expedition period (the first three days of the study) by specialists with a medical degree. A comprehensive assessment of the functional state of the expedition members was carried out twice a day (morning and evening) during the entire 20-day expedition period.

In order to test the first hypothesis, we divided all study participants (*n* = 38) into 2 groups depending on the content of vitamin D in their body. Taking into account the measure of the central tendency of the sample in terms of vitamin D content, we considered it optimal to divide the group of participants into two subgroups along the border of 20.0 ng/mL (the minimum value of vitamin D content in participants was 9.9 ng/mL, the maximum was 56.2 ng/mL, the mean vitamin D concentration for the entire sample was 21.8 ng/mL, and the median was 18.7 ng/mL). The first group included the participants with more severe vitamin D deficiency (indicator values up to 20.0 ng/mL), and the second group consisted of the participants whose vitamin D value was above 20 ng/mL. As a result, 22 people were included in the first group and 16 in the second.

Dividing the sample into groups with and without deficiency of vitamin D in the blood was not possible due to the fact that, in the study, most of the participants had vitamin D deficiency. The participants were selected for the expedition based on their research competence and the uniqueness of the proposed projects and not on other criteria. Along this line, it was decided to divide the groups statistically into those who had this deficiency to a greater and lesser extent, in order to see the differences in the dynamics of the functional states of these participants.

The groups were compared by all parameters (objective, subjective, and projective) measured on the first and last day of the study as well as by individual average values of the parameters for the entire period. To compare the samples, a non-parametric Mann–Whitney U-test was used.

In order to test the second hypothesis, a correlation analysis was carried out with the calculation of Spearman’s correlation coefficients between the level of vitamin D in blood and individual indicators of the dynamics of functional states among the expedition members. These individual trend scores were calculated for each participant based on the overall trend of observations (twice a day, daily) as follows. For each member of the expedition, for each parameter, a regression equation was constructed with the calculation of a coefficient that reflects the average increase in the indicator per unit of time. Graphically, this ratio shows the slope of the trend line. Thus, the larger the coefficient, the higher the growth (with positive values) or the preservation (with negative values) of the indicator in the dynamics of observation. As an example, let us consider the dynamics of the sensorimotor reaction time indicator (CVMR-35) over a 20-day period (Figure 1).

Statistical data processing was carried out using the SPSS 23.00 software package and the Microsoft Excel software package. The following statistical methods were applied: descriptive statistics, exploratory analysis, comparative analysis using the non-parametric Mann–Whitney U-test, and correlation analysis using the Spearman rank correlation coefficient.

## 3. Results

### 3.1. Vitamin D Status among Expedition Members

We studied 38 serum samples taken from the expedition team (women—20 [52.6%], men—18 [47.4%]) from 18 cities of the Russian Federation.

The minimum value was 9.9 ng/mL, the maximum was56.2 ng/mL, the mean concentration of vitamin D in the entire sample was 21.8 ng/mL, and the median was18.7 ng/mL. Figure 2 shows the distribution of the values of this parameter in the sample.

Vitamin D deficiency was found in 2 participants (4%); 26 participants (52%) had insufficient levels; 20 participants (40%) had optimal vitamin D levels; and 2 participants had vitamin D concentrations greater than 50 ng/mL (4%), which is considered to be above norm. All data obtained for the level of 25-hydroxy (calciferol) in serum and the ratio with the participants’ sexes are summarized in Table 2.

According to the processing of participants’ questionnaires, a statistically significant relationship between the concentration of vitamin D in serum and the intake of vitamin–mineral complexes was not found. According to the questionnaire, 11 out of 20 participants with an optimal level of vitamin D in their blood consumed multivitamin supplements containing vitamin D regularly. Only 2 participants (4%) out of 26 participants who were found to be vitamin D deficient mentioned irregular vitamin D consumption in the questionnaire. Two participants were found to be vitamin D deficient. All of them, according to the questionnaires, did not consume vitamin D. Two participants had a higher-than-normal level of vitamin D (≥50 ng/mL); according to the questionnaire, they consumed vitamin D on a regular basis.

The findings suggest that 26% of women and 30% of men are deficient in vitamin D. In women and men, the median concentration of vitamin D was only 1% lower (18.4 ng/mL and 18.6 ng/mL, respectively).

According to the classification, the United Nations distinguishes the following age groups: 15–24; 25–34; 35–44; 45–54; 55–64; 65–74; and 75+ years old [52]. In our study, the minimum age of respondents was 18 years old, which was due to a number of reasons: (1) in Russia, the age of majority is 18, and participation in research does not require the consent of a parent or a legal representative; (2) a criterion for participation in the expedition was an age of 18 years or older. Therefore, all study participants were divided into 6 age groups (ten-year groups): 18–24; 25–34; 35–44; 45–54; 55–64; and 65–74. There were no participants older than 74 years in the study. The distribution by age group is shown in Figure 3.

The study was dominated by participants from the 18–24 and 25–34 age groups, which constituted 34.2% and 31.6%, respectively, while the remaining 4 groups amounted to 34.2%.

In the age group of 18–24 years, the range of vitamin D concentration in blood se-rum was 9.9–25.0 ng/mL. More than 50% of participants in this age group had an insufficient level of vitamin D. Two participants had vitamin D concentration below 12 ng/mL, which corresponds to severe deficiency. The rest of the participants in this age group had an optimal level of vitamin D in serum. The median concentration of vitamin D for this age group was 15.2 ng/mL. In men, the median concentration was 14% higher than in women (17.8 ng/mL and 15.2 ng/mL, respectively).

In the group of participants aged 25–34 years, the range of vitamin D concentration in the blood was 12.2–35.7 ng/mL. More than half of the participants had an optimal level in vitamin D, the remaining participants were deficient in this vitamin (range 12–20 ng/mL). The median concentration in men was also higher than in women by 8%, amounting to 20.9 ng/mL and 19.2 ng/mL, respectively.

In the 35–44 age group, the range of serum vitamin D concentration was 17.3 to 56.2 ng/mL. Half of the participants of this age group had insufficient levels of vitamin D in the blood, and 25% of this group had an optimal level. Two participants of this age group had vitamin D concentrations ≥50 ng/mL. The median concentration in this age group was 36.6 ng/mL.

In the 45–54-year-old group, the range of vitamin D concentration in the blood was 12.3–34.7 ng/mL. Only 44% of the participants had an optimal level of vitamin D; the remaining participants in this group were found to be deficient. The median vitamin D concentration was 21.1 ng/mL. The median concentration of vitamin D in women was more than 2 times higher than that in men: 39.3 and 17.3 ng/mL, respectively.

The 55–64 age group was represented by two participants. The man had a 12.9 ng/mL concentration of vitamin D, which corresponds to deficiency. The woman had an optimal vitamin D concentration—43.9 ng/mL.

In the group aged 65–74 years, the concentration of vitamin D was 25.9 ng/ml, which is the optimal level.

### 3.2. A Comparative Analysis of Objective Indicators of the Functional State of the Participants with Different Levels of Vitamin D in Blood

In order to characterize the heart rate variability of the two groups, we conducted a series of comparative analyses. The groups were compared according to the objective parameters of the VCM. For comparison, measurements were used on the first and the last day of the study, as well as the average values for the parameters of this technique. Thus, according to the integral indicator “functional state level”, no significant differences were found between the groups. Statistically significant differences between the groups with different levels of vitamin D were identified on the first day of the study (morning measurement) in terms of the average duration of RR intervals between sinus contractions (RRNN) and the standard deviation of the duration of RR intervals between sinus contractions (SDNN). The results are shown in Table 3.

According to the data in Table 2, on the first day of the study (morning measurement), the average length of the RR intervals between sinus beats (RRNN) was higher in the participants with a more pronounced vitamin D level than in the participants with a deficiency (*p* = 0.05). The groups also differed in terms of the standard deviation of the duration of RR intervals between sinus beats (SDNN). At the beginning of the expedition (the first day of the study), SDNN indicators in members of the first group with severe vitamin D deficiency were not only lower (38.77 ± 6385 ms) than in members of the second group, but they were even below the norm (Figure 4).

Decreased SDNN values in group 1 with vitamin D deficiency reflect the presence of a pronounced tension in regulatory systems. The adaptive capabilities of the organs of these participants are provided at a higher level of tension in regulatory systems, which leads to increased consumption of the functional reserves of the body. In the second group, as a whole, the values for this parameter (71.88 ± 17.649 ms) corresponded to the norm and did not fall below the median value of the indicator values in the first group.

Thus, significant differences in the functional resources of the organs of the expedition participants were revealed objectively (based on the results of vitamin D content in blood). At the next stage of the study, a comparative analysis of the positive and negative characteristics of the functional states of expedition participants with deficient or optimal levels of vitamin D in the blood was carried out.

In order to characterize the operator performance of the participants in the two groups, we conducted a similar series of comparative analyses according to the Mann–Whitney test. The groups were compared in terms of the quantitative and integral parameters of CVMR-35. There were no significant differences between the two groups in any of the declared parameters.

### 3.3. Comparative Analysis of Subjective and Projective Indicators of the Functional State of Participants with Different Vitamin D Levels

We conducted a series of comparative analyses in order to assess the functional states of the participants in the two groups in terms of vitamin D levels based on the characteristics of “well-being”, “activity”, and “mood” (WAM method). According to the results of the analysis, no statistically significant differences were found in these parameters on the 1st and 20th days of the study (beginning and end of the expedition), and no significant differences were found in the average values of these characteristics. This means that the perceived self-assessment of the functional states of the participants does not differ significantly in groups with different levels of vitamin D.

Also, no statistically significant differences were found between the participants of the two groups in terms of projective indicators (M. Luscher color test) in the dynamics of morning and evening measurements on the first and last days of the study and in the average values of the participants for these parameters.

Thus, the first hypothesis of the study was partially confirmed. According to the positive characteristics of the functional states of participants with different levels of vitamin D in blood, measured by objective, projective, and subjective methods, no statistically significant differences were found. Statistically significant differences were found in the negative characteristics of functional states (according to the objective data of the ECM method): participants with severe vitamin D deficiency on the first day of the study (morning measurement) had a low standard deviation of the duration of RR intervals between sinus contractions (SDNN), which indicates the presence of tension in the regulatory mechanisms.

### 3.4. The Relationship between the Degree of Vitamin D Content in the Blood of the Participants in a Sea Expedition and Their Functional State Indicators

In order to test the second hypothesis, a correlation analysis was carried out. Since the data on the blood vitamin D levels of participants were not normally distributed, Spearman’s rank correlation coefficient was used to analyze the relationships. As a result of the correlation analysis in terms of vitamin D content in the blood of participants, statistically significant correlations with individual coefficients of the dynamics of observations were revealed by objective (hardware) and projective (unconscious) methods. The analysis included the indicators of the first and last days of the study, which reflect the initial (at the start of the expedition) and final (at the final stage of the voyage) levels of psychophysiological resources, as well as the average increases in the main characteristics of functional states during the entire expedition (in the morning and evening measurements). Figure 5 shows the corresponding correlation pleiad.

As can be seen from the data in Figure 5, direct links (moderate in strength) were revealed between vitamin D content and objective indicators that characterize heart rate variability: the SDNN indicator on the 1st day of the study (morning and evening measurements) and the average duration of RR intervals (RRNN) on the 19th day of the study (evening measurement). Such a relationship may indicate that with reduced values of vitamin D in the blood of participants, disturbances in heart rate variability may become more pronounced against the background of an increase in the sympathetic division of the autonomic nervous system and tension and overstrain of regulatory mechanisms. Thus, a pronounced deficiency in vitamin D may be one of the markers of a decrease in the regulatory capabilities of the cardiovascular system.

Assessing the relationship between the indicator of vitamin D content and objective performance indicators, the following patterns should be noted. The shorter the average reaction time (ms), the faster a participant performed a complex visual–motor test, that is, the higher the speed of his sensorimotor reaction. A moderate inverse relationship was found between the vitamin D level in blood and the increase in the indicator, reflecting the average reaction time (ms) in the morning dynamics and in the dynamics of the entire study period, and this may indicate that the response time to sensory stimuli may increase with reduced vitamin D levels. That is, vitamin D deficiency can negatively affect the speed of reactions to stimuli. This is also evidenced by a direct relationship between the vitamin D index and an increase in the integral objective indicator “performance level” during the study period.

The vitamin D content indicator had a moderate direct relationship with the increase in the projective indicator of working capacity in the dynamics of evening measurements. This means that the higher vitamin D content, the easier the participants maintain psychological performance, measured at an unconscious level. The negative projective characteristic of the state of stress demonstrated a negative relationship in support of our hypothesis. The lower the level of vitamin D, the faster the stress increased when working in the extreme climatic and geographical conditions of the Arctic.

## 4. Discussion

Vitamin D deficiency and insufficiency in serum was detected in more than 50% of study participants. The median was 18.7 ng/mL, which corresponds to a deficient state. The optimal level of 25-hydroxy (calciferol) was more typical for women and amounted to 18% in the female sample, while in the male sample, only 22% of the participants had an optimal level. The medians were almost equal and amounted to 18.4 ng/mL and 18.6 ng/mL for women and men, respectively. It should be noted that the sample included 20 women and 18 men, so the percentage is not presented for the entire sample as a whole but rather in accordance with gender. In a study by Kondratiev et al. conducted in St. Petersburg, vitamin D levels in men and women were almost at the same level; no statistically significant differences were found. The general trends indicated lower concentrations of vitamin D in men, regardless of season and age. In women, the lowest vitamin D content was observed in old age, regardless of the season of the year [53,54,55]. For future research, we see the need to conduct a more detailed survey of participants about vitamin D intake and other factors which may be associated with Vitamin D levels in human blood. In particular, a questionnaire should disclose information about the dosage of vitamin D intake, about health status (e.g., list of chronic diseases), and about specificities of work and lifestyle.

Our study revealed and described significant relationships between the nature of the dynamics of objective and projective indicators of the functional state of expedition members with the content of vitamin D in their blood. The differences between objective, projective, and subjective indicators of the functional states in the groups with and without severe vitamin D deficiency were also described.

Direct links were found between the level of vitamin D and objective indicators that characterize heart rate variability. The average duration of RR intervals between sinus beats was longer in the participants with vitamin D levels above 20 ng/mL than in the participants with a severe deficiency. SDNN scores in vitamin D deficient participants were almost twice as low as those in the second group. This is supported by Li Ye Chen et al., who found that vitamin D deficiency is associated with a lower SDNN compared to that of the sufficient group. The reason may be that vitamin D deficiency is associated with a decrease in the functions of the parasympathetic system [56]. Canpolat et al.’s studies have shown that autonomic function of the heart can be impaired in the absence of heart damage and symptoms in people with vitamin D deficiency [57].

At the same time, a number of researchers have found that there is no relationship between heart rate variability and vitamin D content in the blood. Nalbant A. et al. found that heart rate variability did not differ significantly in patients with low vitamin D status [58]. Matheï C. et al. found no significant relationship between vitamin D deficiency and physical performance in the elderly [59]. Arslan D. et al. found no relationship between vitamin D levels, functional status, and quality of life in patients with osteoarthritis [60].

Further research should expand the assessment of factors that may increase the negative impact of vitamin D deficiency on the circulatory system. These factors include metabolic syndrome. A number of authors describe various mechanisms and effects of vitamin D deficiency on the development of cardiovascular diseases and metabolic syndrome [61,62,63].

It is worth noting that the relatively small number of similarly designed studies combined with small sample sizes makes it difficult to compare and generalize the results of this study with those of other expeditions. At the same time, the data obtained contribute to the discussion on the mutual influence of vitamin D content in blood and the functioning of the human cardiovascular system. The question of the relationship between vitamin D content and performance and stress indicators is also subject to discussion.

Due to the fact that vitamin D deficiency was determined in more than 85% of the subjects included in the study, it is necessary to include its definition in the list of possible additional tests that are carried out as part of the medical examination of participants prior to the expedition. This will make it possible to determine vitamin D deficiency in a timely manner and take measures to replenish it before and during the expedition, which will increase the adaptive capabilities of the participants in Arctic expeditions.

A relatively small sample size could be a possible limitation of this study. Larger samples collected during future expeditions in other regions could help us to study this problem in more depth.

## 5. Conclusions

The assessment of the functional state was carried out according to different indicators: objective, subjective, and projective. At the same time, significant differences between the groups with different vitamin D levels were revealed only in terms of objective parameters of expedition members’ functional states. Hypothesis 1 has been partially confirmed. At the beginning of the expedition, it was found that the functional state of participants with more severe vitamin D deficiency was characterized by a shorter average duration of RR intervals (*p* = 0.050) and reduced SDNN values (*p* = 0.015), which indicates a higher level of tension in regulatory mechanisms.

According to the results of the correlation analysis, the most pronounced differences in the nature of the dynamic series of the parameters of the operator’s performance (indicators of speed and average response time) were most clearly manifested. It was found that with reduced vitamin D levels, the performance of visual–motor tasks by the participants tended to deteriorate from the beginning to the end of the expedition (*r* = 0.510). A relationship was found between the indicators of vitamin D in the blood of the participants and an average increase in the projective indicator of working capacity in the dynamics of evening measurements (*r* = 0.485) and an inverse relationship with an increase in the projective indicator of stress in the dynamics of morning measurements (*r* = −0.334).

With an increase in the severity of vitamin D deficiency in the blood, the adaptive capabilities of participants decrease during an expedition to the Arctic. The obtained results and conclusions make it possible to expand the possibilities for assessing the psychological risks of failures in adaptation to professional activities in the Arctic as well as to develop practical recommendations.

## Figures and Tables

**Figure 1 ijerph-20-06092-f001:**
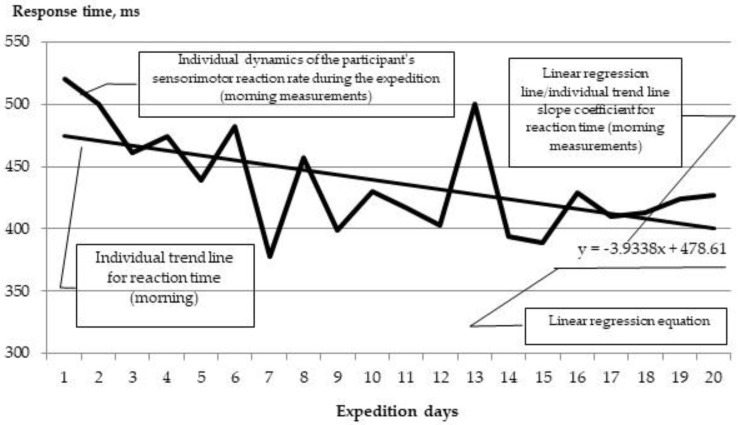
Dynamics of the reaction time indicator of the research participant during the expedition period.

**Figure 2 ijerph-20-06092-f002:**
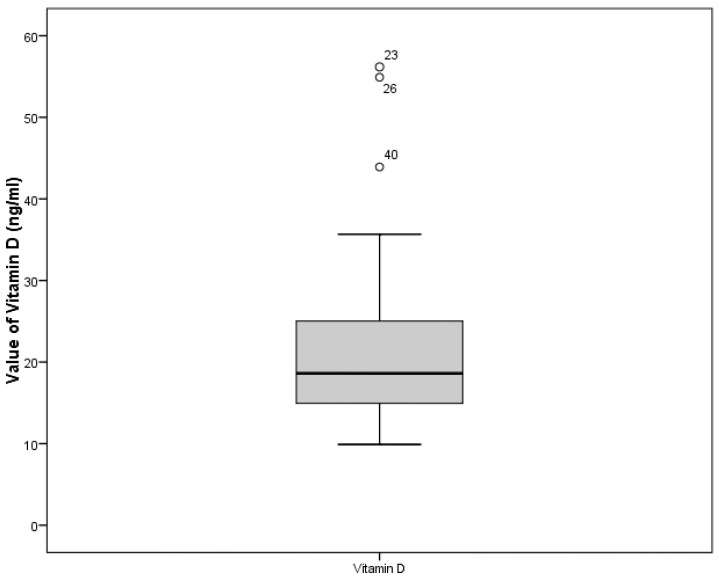
Sample distribution by vitamin D content. Note: Data outliers reflect those participants who were found to have higher optimal vitamin D levels (values are 56.2 ng/mL; 54.9 ng/mL; 43.9 ng/mL).

**Figure 3 ijerph-20-06092-f003:**
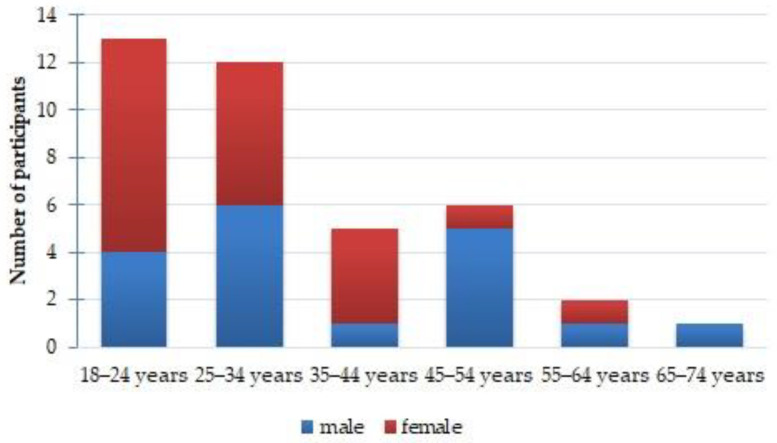
Distribution of expedition members by age and gender.

**Figure 4 ijerph-20-06092-f004:**
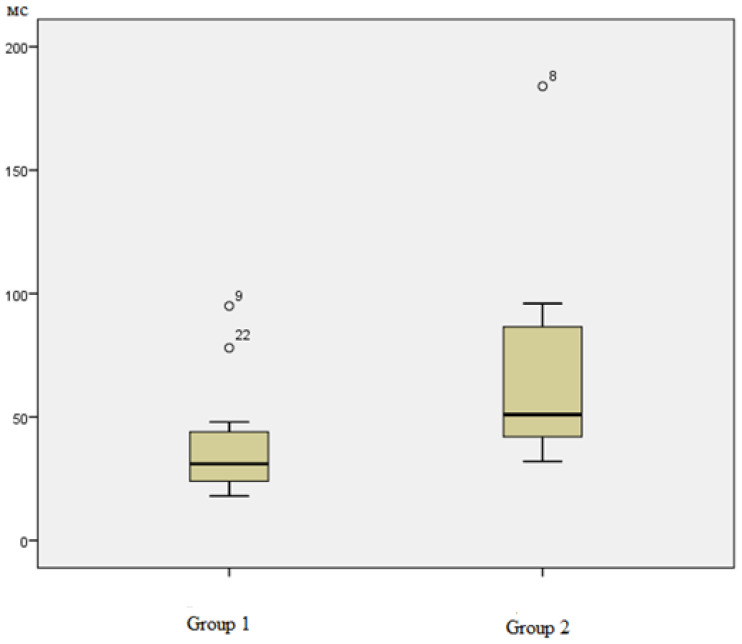
Diagrams of the distribution of RMS/SDNN parameter values in the first and second groups (day 1, morning measurement).

**Figure 5 ijerph-20-06092-f005:**
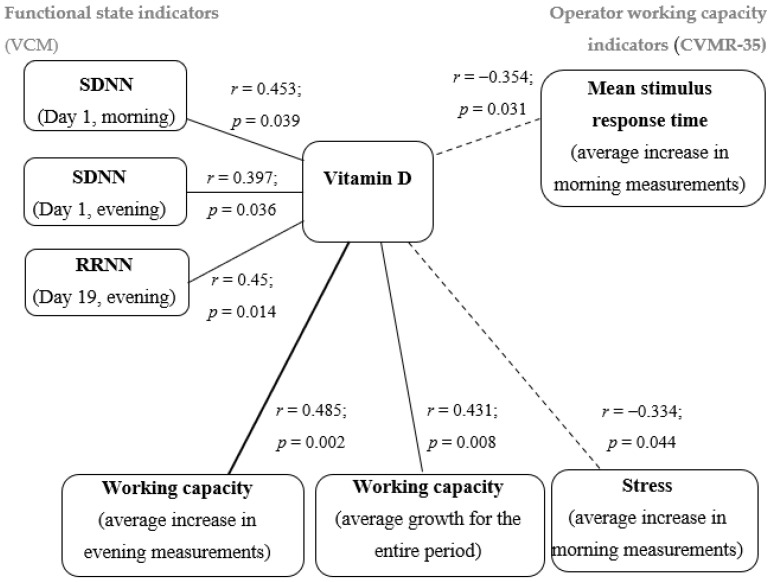
Correlation pleiad of the relationship between vitamin D level in blood and the nature of the dynamics of objective and projective indicators of expedition members’ functional states.

**Table 1 ijerph-20-06092-t001:** Positive and negative characteristics of the functional states of the expedition members.

Method	Positive Characteristics of the Functional State	Negative Characteristics of the Functional State
VCM	optimal and close to optimal level of the general functional state of the body	from the permissible and critical levels of the general functional state of the organism
CVMR-35	medium and high levels of operator working capacity	reduced and low levels of operator working capacity
M. Luscher’s technique	medium and high working capacity	stressful state, unproductive neuromuscular activity (SO), reduced and low working capacity
WAM methodology	medium and high levels of well-being, activity, and mood	reduced and low levels of well-being, activity, and mood

**Table 2 ijerph-20-06092-t002:** Vitamin D levels in expedition members.

Gender	Percentage of the Total Number of Men or Women			
	Above the Norm	Optimal Vitamin D Level	Insufficient Level of Vitamin D	Vitamin D Deficiency
Women	4%	18%	22%	4%
Men		22%	30%	

**Table 3 ijerph-20-06092-t003:** Results of a comparative analysis of objective indicators of the functional state (VCM) in groups with different levels of vitamin D in the blood of the expedition members.

Parameter	Group 1 (Vitamin D Value to 20 ng/mL) M ± SE	Group 2 (Vitamin D Values above 20 ng/mL) M ± SE	Mann–Whitney U-Test Value	Significance of Differences (2-Tailed) p	Norms
RRNN	715.00 ± 27.607	812.13 ± 38.433	25.000	0.050	667–1000 Normocardia
(Day 1, morning freeze)	38.77 ± 6.385	71.88 ± 17.649	18.500	0.015	40–80 мc

Note: The table presents the values only for the parameters with significant differences.

## Data Availability

Certificate of registration of the database 2022621401, 7 June 2022. Application No. 2022621296 dated 7 June 2022. Dynamics of functional states and psychological adaptation of the participants of the marine expedition to the Arctic region.

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
