# Peer review of "The Functional States of the Participants of a Marine Arctic Expedition with Different Levels of Vitamin D in Blood"

_ijerph, 2023, doi:10.3390/ijerph20126092_

Round 1

Reviewer 1 Report

1.      Authors mention

Vitamin D sufficiency was assessed on the basis of the following criteria: vitamin D content within 30–80 ng/ml was considered normal, the range of 20–30 ng/ml corresponded to deficiency, 10–19 ng/ml -- to shortage and values less than 10 ng/ml -- to severe shortage (36).

Reference 36 mentions

Thus, based on these and other studies, it has been suggested that vitamin D deficiency be defined as a 25(OH)D below 20 ng/ml, insufficiency as a 25(OH)D of 21–29 ng/ml, and sufficiency as a 25(OH)D of 30–100 ng/ml. 

Since there is disparity, use of shortage term and also unit mentioned needs correction.

Definition of Vitamin D deficiency: the values used need justification
As per the Fact Sheet for Health Professionals by NIH is different, this authors may refer to
(https://ods.od.nih.gov/factsheets/VitaminD-HealthProfessional/)

2.      Plagiarism similarity index is high 26%

To be brought down as per journals guidelines for original articles  

3.      Abstract and conclusion should clearly mention differences found in groups.

Clearly mentioning results in smaller sentences required

Author Response

Dear Reviewer,
Thank you very much for your interest and your time devoted to our work! We have carefully studied your recommendations and corrected the article. Significant changes to the text are marked in red for easy identification.

1. Vitamin D sufficiency was assessed on the basis of the following criteria: vitamin D content within 30–80 ng/ml was considered normal, the range of 20–30 ng/ml corresponded to deficiency, 10–19 ng/ml -- to shortage and values less than 10 ng/ml -- to severe shortage (36).

Reference 36 mentions

Thus, based on these and other studies, it has been suggested that vitamin D deficiency be defined as a 25(OH)D below 20 ng/ml, insufficiency as a 25(OH)D of 21–29 ng/ml, and sufficiency as a 25(OH)D of 30–100 ng/ml. 

Since there is disparity, use of shortage term and also unit mentioned needs correction.

Definition of Vitamin D deficiency: the values used need justification
As per the Fact Sheet for Health Professionals by NIH is different, this authors may refer to
(https://ods.od.nih.gov/factsheets/VitaminD-HealthProfessional/)

Thank you for your useful comments and the link to the Fact Sheet for Health Professionals by NIH. We have rewritten the methods, results and discussion sections in accordance with your recommendations.

We also made additional clarifications (lines 303-319).

  1. Plagiarism similarity index is high 26%

To be brought down as per journals guidelines for original articles  

Thank you very much for your comment! We have changed the text of the article and improved the % originality.

  1. Abstract and conclusion should clearly mention differences found in groups.

We have made changes to the abstract and conclusion.

Best regards, the authors

Reviewer 2 Report

The study explores an interesting topic in a vulnerable population, i.e., the vitamin D deficiency in Arctic explorers. The results have pragmatical implications, therefore the research could be useful for health specialists involved in monitoring participants in such expeditions. Several formal and structural aspects need to be addressed, in order to increase the clarity and the impact of this article, please see below:

-the methodology includes several instruments with limited availability in the Western literature, therefore consider describing psychometric properties for the self-assessment questionnaire or validation studies for psychophysiological methods, if available; in the absence of such reference points, the reader will face difficulties in understanding the results of the current study;

-consider defining within the abstract the strength of the correlation between the evaluated variables;

-insert reference for Khasnulin et al. In line 52, because references 13 and 14 are not about this researcher;

-consider inserting reference(s) for lines 56-60; the same for the study of Bulgarian scientists, mentioned  in lines 64-65 and for the other study mentioned in lines 67-70; also, please rephrase that sentence, because „in another study... was studied” does not seem right;

-line 72- „the severity of the environment” seems incomplete; please detail or rephrase;

-line 79- which country?

-line 91- please insert reference for Baevsky’s study;

-the formulation of the study’s objective in lines 116-117 is ambiguous, please rephrase for clarity;

-maybe consider using past tense for the hypotheses in lines 123 and 126, because the study was already done;

-lines 148-149- the sentence starting with „It was...” should be rephrased for clarity;

-line 169- why is „centrifugation” highlighted in yellow?

-lines 174-175- please rephrase, there is no verb in that sentence;

-lines 275-276- please insert reference for Aminev’s study;

-figure 4- consider removing the Russian characters, the English labels are sufficient;

-line 461- „the first hypothesis of the study was partially confirmed” vs. line 551- „Hypothesis 1 was confirmed”;

-line 150- what biological parameters and psychological variables were included in the preliminary check-up? Vitamin D deficiency was determined in more than 85% of the subjects enrolled in the study, so maybe one of the study’s conclusions could be to include some supplementary analyses in the check-up for this population, especially if functional consequences exist in individuals with vitamin D deficiency.

The overall quality of the English language is fair. There are some isolated syntactical aspects that have been mentioned in the above commentaries.

Author Response

Dear Reviewer,
Thank you very much for your interest and your time devoted to our work! We have carefully studied your recommendations and corrected the article. Significant changes to the text are marked in red for easy identification.

-the methodology includes several instruments with limited availability in the Western literature, therefore consider describing psychometric properties for the self-assessment questionnaire or validation studies for psychophysiological methods, if available; in the absence of such reference points, the reader will face difficulties in understanding the results of the current study;

Thank you for your comment! Added in lines 223-244.

We also clarify that during certification and registration of the device, the validity of the reliability of the test results is checked. The results of testing on this device correlate with the results performed on other psychophysiological devices.

-consider defining within the abstract the strength of the correlation between the evaluated variables;

Added

-insert reference for Khasnulin et al. In line 52, because references 13 and 14 are not about this researcher;

These references refer to Selye's concept, which is referred to in this proposal. For this reason, these links have been left.

-consider inserting reference(s) for lines 56-60;

Added

the same for the study of Bulgarian scientists, mentioned  in lines 64-65 and for the other study mentioned in lines 67-70;

Because both sentences referred to the same source, the link was placed on the second sentence, now it has been added to the first sentence as well

also, please rephrase that sentence, because „in another study... was studied” does not seem right;

Changed to:

The dynamics of stress and recovery reactions and their relationship with the perception of environmental development during one year of polar wintering of expedition members in various environmental conditions of the subantarctic and Antarctic polar stations was studied by the international scientific team

-line 72- „the severity of the environment” seems incomplete; please detail or rephrase;

Changed and added

«…on the degrees of environmental extremeness (the authors found that the stress response duration is proportional to the severity of environmental conditions: it is longer in polar conditions than in subantarctic ones).»

-line 79- which country?

Added «…by Russian scientists»

-line 91- please insert reference for Baevsky’s study;

Added

-the formulation of the study’s objective in lines 116-117 is ambiguous, please rephrase for clarity;

Changed to:

«….to identify and describe the relationship between changes in objective psycho-physiological and subjective psychological parameters of functional states and the lev-el of vitamin D deficiency in the blood of participants in a sea expedition to the Arctic region during a 20-day trip.»

-maybe consider using past tense for the hypotheses in lines 123 and 126, because the study was already done;

Changed

-lines 148-149- the sentence starting with „It was...” should be rephrased for clarity;

Changed

-line 169- why is „centrifugation” highlighted in yellow?

This is a mistake, removed, sorry

-lines 174-175- please rephrase, there is no verb in that sentence;

Changed

«…Liquid chromatograph Agilent 1200 (USA), mass spectrometer AB Sciex 3200 MD (Singapore) were used for the study.»

-lines 275-276- please insert reference for Aminev’s study;

Added

-figure 4- consider removing the Russian characters, the English labels are sufficient;

Changed

-line 461- „the first hypothesis of the study was partially confirmed” vs. line 551- „Hypothesis 1 was confirmed”;

Corrected

Hypothesis 1 has been partially confirmed

-line 150- what biological parameters and psychological variables were included in the preliminary check-up?

Changed

An examination by a general practitioner was a prerequisite for participation in sea expeditions which was to identify contraindications to the relevant working conditions.

Vitamin D deficiency was determined in more than 85% of the subjects enrolled in the study, so maybe one of the study’s conclusions could be to include some supplementary analyses in the check-up for this population, especially if functional consequences exist in individuals with vitamin D deficiency.

In future research we see the need to conduct a more detailed survey of participants about vitamin D intake and other factors, which may be associated with Vitamin D levels in human blood. In particular, a questionnaire should disclose information about the dosage of vitamin D intake, about health status (e.g., list of chronic diseases, which can affect the status of vitamins in the body), about specificity of work and lifestyle.

Added in conclusion

Due to the fact that Vitamin D deficiency was determined in more than 85% of the subjects included in the study, it is necessary to include its definition in the list of possible additional tests that are carried out as part of the medical examination of participants prior to the expedition. This will make it possible to timely determine vitamin D deficiency and take measures to replenish it before and during the expedition, which will increase the adaptive capabilities of the participants in the Arctic expeditions.

Best regards, the authors

Reviewer 3 Report

The authors have addressed the significance of vitamin D on the functional states of participants in a marine Arctic expedition in this pilot study. They have made a noteworthy effort to measure the participants' vitamin D levels, evaluate their functional states, and provide a detailed explanation of the methods and results. I have the following concerns, 

Small sample size: As authors have also understood/realized it and mentioned it as a limitation in the discussion section. This work still has its merit in showing the association between vitamin D and functional states. Because of the poor sample size and distribution, the authors have grouped the participants with <20 ng/ml vitamin D as group 1 and >20 ng/ml as group 2, which is not ideal. A normal level is typically 30 ng/ml or higher. 

Heart rate variability (HRV): Please detail the methods of how the HRV was measured, as the environmental factors can easily increase the variability of HRV and the software used to analyze the data. Also, explain why authors have not measured or reported the respiratory rate in these participants. Other confounding factors that alter HRV include sex, body weight, menstrual cycle, and disease status should be addressed. 

Sample distribution: As more than 70% of participants are in the 18-24 age group - what is the association between vitamin D and functional states in this population?

Methods: Please write only methods and do not explain each measurement in the methods section. 

Discussion: Please add additional studies that have found no link between vitamin D and functional states or other outcomes. The vitamin D research field has conflicting data, so it would be helpful to discuss more studies that support both sides. 

Author Response

Dear Reviewer,
Thank you very much for your interest and your time devoted to our work! We have carefully studied your recommendations and corrected the article. Significant changes to the text are marked in red for easy identification.

  1. Small sample size: As authors have also understood/realized it and mentioned it as a limitation in the discussion section. This work still has its merit in showing the association between vitamin D and functional states. Because of the poor sample size and distribution, the authors have grouped the participants with <20 ng/ml vitamin D as group 1 and >20 ng/ml as group 2, which is not ideal. A normal level is typically 30 ng/ml or higher. 

As you have justifiably noted, the vitamin D research field contain conflicting data. For example, the Fact Sheet for Health Professionals by NIH (https://ods.od.nih.gov/factsheets/VitaminD-HealthProfessional/ indicates that the normal) level is typically 20 ng/ml or higher, while Holick et al. research indicate that the normal level is from 30 ng/ml or higher. Due to another reviewer's recommendation and the small sample amount, all results we decided to revise our manuscript in accordance with the classification mentioned above.

We also made additional clarifications (lines 303-319).

  1. Heart rate variability (HRV): Please detail the methods of how the HRV was measured, as the environmental factors can easily increase the variability of HRV and the software used to analyze the data. Also, explain why authors have not measured or reported the respiratory rate in these participants. Other confounding factors that alter HRV include sex, body weight, menstrual cycle, and disease status should be addressed. 

To assess the HRV, a psychophysiologist device was used as the most mobile, which does not require the location of the research participant on the couch; to perform the test, it is enough to take the device in hand and the execution time is up to 5 minutes. This mobility brought some limitations to the studies and introduced additional variability in the results of the study. It was not possible for two researchers to place 38 members of the expedition on the couch every day in the morning and in the evening and perform the HRV. This device does not imply an assessment of the respiratory rate due to the lack of sensors.

  1. Sample distribution: As more than 70% of participants are in the 18-24 age group - what is the association between vitamin D and functional states in this population?

We have added data on detected vitamin D concentrations in this age group.

We did not perform this analysis because Pearson's correlation coefficient turned out to be quite low - 0.371 at p 0.022.

  1. Methods: Please write only methods and do not explain each measurement in the methods section. 

We shortened and adjusted the description of the methods, taking into account the recommendations of all reviewers.

  1. Discussion: Please add additional studies that have found no link between vitamin D and functional states or other outcomes. The vitamin D research field has conflicting data, so it would be helpful to discuss more studies that support both sides. 

Thanks for the comment! The discussion has been expanded. Lines 548-558.

Best regards, the authors